# Five colour variants of bright luminescent protein for real-time multicolour bioimaging

Kazushi Suzuki[1], Taichi Kimura[2], Hajime Shinoda[1], Guirong Bai[3], Matthew J. Daniels[4], Yoshiyuki Arai[1,2,3], Masahiro Nakano[1,2,3] & Takeharu Nagai[1,2,3]

Luminescence imaging has gained attention as a promising bio-imaging modality in situations where fluorescence imaging cannot be applied. However, wider application to multicolour and dynamic imaging is limited by the lack of bright luminescent proteins with emissions across the visible spectrum. Here we report five new spectral variants of the bright luminescent protein, enhanced Nano-lantern (eNL), made by concatenation of the brightest luciferase, NanoLuc, with various colour hues of fluorescent proteins. eNLs allow five-colour live-cell imaging, as well as detection of single protein complexes and even single molecules. We also develop an eNL-based $Ca^{2+}$ indicator with a 500% signal change, which can image spontaneous $Ca^{2+}$ dynamics in cardiomyocyte and neural cell models. These eNL probes facilitate not only multicolour imaging in living cells but also sensitive imaging of a wide repertoire of proteins, even at very low expression levels.

[1] Department of Biotechnology, Graduate School of Engineering, Osaka University, 2-1 Yamadaoka, Suita 565-0871, Japan. [2] Department of Biotechnology, School of Engineering, Osaka University, 2-1 Yamadaoka, Suita 565-0871, Japan. [3] Department of Biomolecular Science and Engineering, The Institute of Scientific and Industrial Research, Osaka University, 8-1 Mihogaoka, Ibaraki 567-0047, Japan. [4] BHF Centre for Regenerative Medicine, Division of Cardiovascular Medicine, West Wing Level 6, John Radcliffe Hospital, Oxford OX3 9DU, UK. Correspondence and requests for materials should be addressed to T.N. (email: ng1@sanken.osaka-u.ac.jp).

Live-cell imaging with fluorescent proteins (FPs) has become a gold standard of biological imaging with the aid of an expanded FP colour palette. However, the requirement of excitation light for fluorescence detection sometimes causes serious problems such as phototoxicity, perturbation of photo-dependent biological phenomena and autofluorescence from the specimen.

Instead of FPs, using luminescent proteins (LPs), such as luciferase, can circumvent these problems. LPs generate an emission signal by catalysing a confined chemical reaction with a luminescent compound, so they are totally independent from an external light source. However, the output photons from conventional LPs are generally insufficient to provide spatiotemporal resolution equivalent to fluorescence.

To overcome the dim signal limitation of LPs, researchers have used directed evolution to improve the intrinsic properties of luciferase such as its luminescent quantum yield, catalytic turnover and stability[1,2]. Recently, the brightest luciferase, NanoLuc (Nluc), was developed through genetic engineering of Oplophorus luciferase (Oluc), which is derived from the deep-sea shrimp *Oplophorus gracilirostris*, along with its optimal substrate, furimazine[3,4]. Although the intense luminescence of Nluc is very useful for visualizing fast physiological events and proteins with lower expression levels, Nluc application is restricted to single biological events due to the lack of distinct colour variants[5,6]. Therefore, there is a strong demand for colour variants of Nluc to monitor multiple biological events.

Colour variants of luciferase are typically developed by rationally altering residues in the active pocket, which alters the chemical form of the excited luminescent substrate, thereby allowing a colour-shifted emission spectrum[7]. This approach has not proven particularly effective for Nluc owing to the modified chemical structure of furimazine.

An alternative way to develop colour variants of luciferase is to harness Förster resonance energy transfer (FRET) to a FP that emits the colour of interest. Tight concatenation of luciferase with a FP results in luminescence from the acceptor FP owing to efficient FRET[8–10]. For example, colour hues of *Renilla* luciferase variants (Rluc8_S257G and Rluc8.6) were effectively expanded towards yellow and orange by concatenation with Venus (Yellow FP) and mKusabiraOrange2 (Orange FP), yielding yellow (YNL) and orange Nano-lantern (ONL), respectively[9,10]. These facilitate monitoring of multiple cellular events, including the dynamics of subcellular structures, and gene expression[10].

Inspired by the strategy for Nano-lantern development, here, we create colour variants of Nluc with the aid of FRET. The efficiency of FRET generally increases as the linker shortens, whereas large truncation of the linker in FPs substantially affects folding. Thus, we systematically identify the optimal linker between luciferase and FP that confers high FRET efficiency. These colour variants allow five-colour live-cell imaging, as well as detection of single protein complexes and even single molecules. We also develop an enhanced Nano-lantern (eNL)-based $Ca^{2+}$ indicator with a 500% signal change, which can image $Ca^{2+}$ dynamics in cardiomyocytes.

## Results

### Development of multicolour bright LPs.
To develop colour variants of Nluc, we fused Nluc with mTurquoise2 (mTQ2; ref. 11; cyan FP), mNeonGreen[12] (green FP), Venus[13] (yellow FP), mKOκ[14] (orange FP) and tdTomato[15] (red FP) as FRET acceptors. We speculated that the unstructured residues of Nluc and FPs could work as linkers and be deleted without disrupting the function of either protein. First, we made a library of Nluc and FPs with different linker lengths by systematically truncating

the N terminus of Nluc and C terminus of FPs, as well as randomizing two residues at the junction, derived from a KpnI site (Supplementary Figs 1–6). By screening for high FRET efficiency and absolute brightness, we identified colour variants that exhibited the highest FRET efficiency. We named them cyan eNL (CeNL), green eNL (GeNL), yellow eNL (YeNL) and red eNL (ReNL) (Fig. 1a).

However, the FRET efficiency of mKOκ-Nluc pairs turned out to be moderate. Thus, we attempted to insert mKOκ into the loop region of Nluc (Supplementary Fig. 7 and Supplementary Note 1). During screening, we found that a variant with the highest FRET efficiency had an accidental 22-residue insertion in addition to a GGSGGS linker at the N termini of the mKOκ domain, whereas there was no linker at the C termini. We called it 'orange eNL' (OeNL; Fig. 1a and Supplementary Fig. 8).

Next, the luminescent spectrum of eNLs: CeNL, GeNL, YeNL, OeNL and ReNL, was measured with distinct emission spectra, peaking at 475, 520, 530, 565, and 585 nm, respectively (Fig. 1b). To investigate the brightness of eNLs, we performed side-by-side comparisons of the luminescence intensity of the eNLs, Nluc, Nano-lanterns (yellow, YNL; cyan, CNL; orange, ONL) and Rluc8. Interestingly, the luminescence intensity of CeNL and GeNL was greater than that of Nluc by 2.0- and 1.8-fold, respectively (Fig. 1c). Although the brightness of YeNL, OeNL and ReNL was moderate compared with that of Nluc, they were still brighter than Nano-lanterns with corresponding colour by 4- to 1.8-fold. (Fig. 1c).

To investigate how the luminescence intensities of CeNL and GeNL became brighter than NLuc, we compared luminescence quantum yield (LQY) and the enzymatic parameters ($K_m$ and $k_{cat}$) of the eNLs and Nluc (Supplementary Fig. 9 and Supplementary Table 1). $k_{cat}$ of all eNLs were almost identical to that of Nluc, suggesting that FP fusion did not perturb Nluc enzymatic activity. In contrast, the LQY of CeNL and GeNL became larger than that of Nluc, while LQY of others were comparable or less (Supplementary Table 1). These results indicate that the enhancement of luminescence intensities in CeNL or GeNL are due to the enhancement of LQY by means of efficient FRET from Nluc to FPs with high fluorescence quantum yield.

To our surprise, we could detect the luminescence from a single GeNL molecule immobilized on the glass surface (Fig. 2a). The average total photon numbers from individual spots was consistent with the value estimated from the kinetic parameters associated with bulk solution analysis (Fig. 2b and Supplementary Note 2). Moreover, luminescence intensity at each region of interest (ROI) exhibited a stepwise transition between 'bright states' with $75 \pm 30$ photon emission and 'dark states' with emission similar to the background (Fig. 2c). We reasoned that those two states correspond to association and dissociation between Ni-NTA and single GeNL molecules fused with a his-tag, which might occur during the observation times (the reported dissociation half-times of his-tag and Ni-NTA are $\tau_{fast} = 110$ s and $\tau_{slow} = 386$ s, respectively[16]). These results suggested not only the possibility of single-molecule observation but also the potential of eNLs for single-molecule-based superresolution luminescence imaging. The weak binding between GeNL and target molecules with Ni-NTA might enable sub-diffraction imaging, in a similar manner to the 'universal point accumulation imaging in the nanoscale topography'[17] method.

### Application as a fusion tag for subcellular structures.
To demonstrate the use of eNLs as fusion tags for live-cell imaging, we constructed 14 eNL fusion proteins. These fusion proteins were localized appropriately in living cells, including to

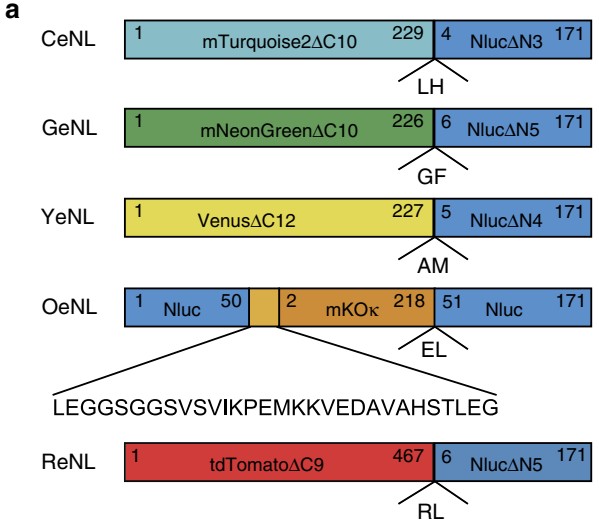

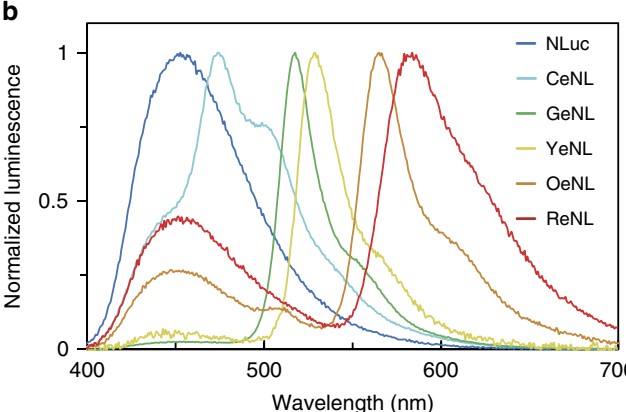

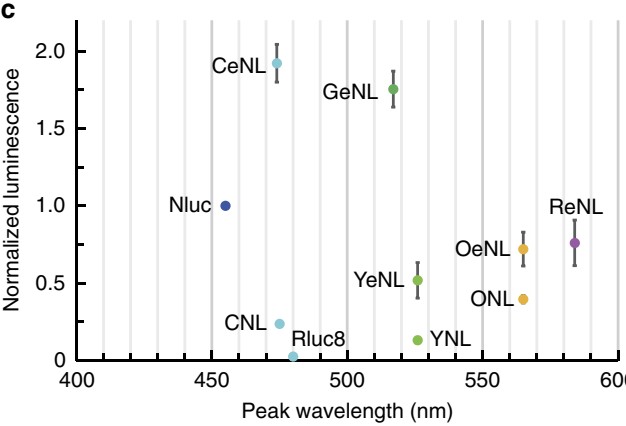

**Figure 1 | Development of multicolour eNLs and their characterization.**
(**a**) Schematic diagram of multicolour eNLs. Numbers represent the relative amino-acid position in the original protein. (**b**) Emission spectra of LPs were measured in triplicate, and representative data are shown. (**c**) Luminescence intensities of equimolar amount of LP on addition of 25 μM substrate, CNL, YNL, ONL, Rluc8, Nluc, GeNL, CeNL, YeNL, OeNL and ReNL. Luminescence intensities were measured in triplicate, and data are presented as means ± s.d.'s.

microtubules, histones and intermediate filaments that require a monomeric character (Fig. 3 and Supplementary Figs 10–13). Notably, CeNL with a clathrin fusion tag demonstrated the perinuclear and punctate plasma membrane distribution corresponding to single clathrin-coated pits, as described previously[18].

According to a crystallographic study, only 180 molecules of the clathrin light chain are incorporated into clathrin-coated pits[19], meaning that eNLs are capable of serving as a fusion tag for visualizing supramolecular complexes that consist of a small number of proteins.

To demonstrate the utility of the eNL colour palette, we co-expressed Nluc and the eNLs in the mitochondria lumen (mito-Nluc), endoplasmic reticulum (ER) lumen (CeNL-ER), nucleolus (GeNL-fibrillarin), inner plasma membrane (Lyn-OeNL) and nucleus (ReNL-H2B). As a result, five different colour luminescence signals were clearly visualized by optical filtering and spectral unmixing (Fig. 4 and Supplementary Fig. 14).

To track the dynamics of multiple subcellular structures, we expressed GeNL and ReNL in lysosome and nucleus by fusion with lysosome-associated membrane protein (LAMP, GeNL is located on the outer surface of lysosome) and histone H2B, respectively, and we tracked their trajectories for 20 min (Supplementary Movie 1). The results demonstrate that eNLs are sufficient for examining the dynamics of multiple subcellular structures simultaneously.

**Development of Ca$^{2+}$ indicators based on GeNL.** To expand the application of eNLs to biosensing, we sought to develop a Ca$^{2+}$ indicator based on GeNL. We inserted a fusion protein of calmodulin and M13 (CaM–M13) into Nluc, in which the conformational change of CaM–M13 by Ca$^{2+}$-binding induces the reconstitution of the split Nluc moiety[20] (Fig. 5a,b). We found that insertion of CaM–M13 between Gly$_{66}$ and Leu$_{67}$ of the Nluc moiety could work as Ca$^{2+}$ indicator (Supplementary Note 3 and Supplementary Fig. 15). After two rounds of directed evolution by error-prone PCR, we obtained a construct with three mutations (K30R, E114V and V142E) in the CaM domain, and a signal change on Ca$^{2+}$ binding of 500%. The $K_d$ value for Ca$^{2+}$ of this construct was 480 nM. Thus, we named it GeNL(Ca$^{2+}$)_480 (Fig. 5a,c,d). The Ca$^{2+}$ affinities of GeNL(Ca$^{2+}$) could be tuned from 60 to 520 nM by changing the linker length between CaM mutants and M13, as reported previously[21] (Supplementary Table 2).

To demonstrate the performance of GeNL(Ca$^{2+}$), we observed Ca$^{2+}$ dynamics induced by histamine in HeLa cells. On addition of 10 μM histamine, an acute Ca$^{2+}$ spike followed by Ca$^{2+}$ oscillations with smaller amplitudes was detected at 30 Hz (Fig. 6a,b and Supplementary Movie 2). Furthermore, we were able to observe the propagation of a cytoplasmic Ca$^{2+}$ wave from one end of the cell to the other (Fig. 6c).

We also made side-by-side comparisons of its performance with the benchmark Ca$^{2+}$ indicators, GCaMP3 (ref. 22) and Fura-2 (ref. 23), in GH3 cells (rat pituitary tumour), which show spontaneous Ca$^{2+}$ spikes (Supplementary Fig. 16). All indicators produced a dynamic signal trace. The signal to noise ratio of GeNL(Ca$^{2+}$)_480 (SNR 120 ± 12 at 1.3 Hz frame rate and 74 ± 3.0 at 33 Hz frame rate; data are presented as mean ± s.d.; $n = 6$ cells) was superior to that of Fura-2 (SNR 9.6 ± 0.71 at 1.3 Hz frame rate; $n = 6$ cells), but inferior to that of GCaMP3 (SNR 590 ± 25 at 33 Hz frame rate; $n = 6$ cells). We also detected spontaneous Ca$^{2+}$ spikes in GH3 cells expressing GeNL (Ca$^{2+}$)_480 over 20 min. In contrast, the signals from Fura-2 were severely diminished 10 min after starting of observation due to phototoxic and photobleaching effects.

Recently, cardiomyocytes derived from human-induced pluripotent stem cells (hiPSC-CMs) provide a platform for personalized drug screening in vitro. Several fluorescent indicators have been used to detect drug effects on cardiomyocyte-beating behaviour[24,25]. To test the applicability of GeNL(Ca$^{2+}$), we expressed GeNL(Ca$^{2+}$)_520 in cardiomyocytes by means

of an adeno-associated virus (AAV) infection system. GeNL(Ca$^{2+}$)_520 was imaged at 60 Hz revealing a periodic luminescence change synchronized with cardiomyocyte

contraction for 35 min (Fig. 7a and Supplementary Movie 3). When cardiomyocytes were treated with astemizole, which perturbs beat behaviour by off-target human Ether- à-go-go

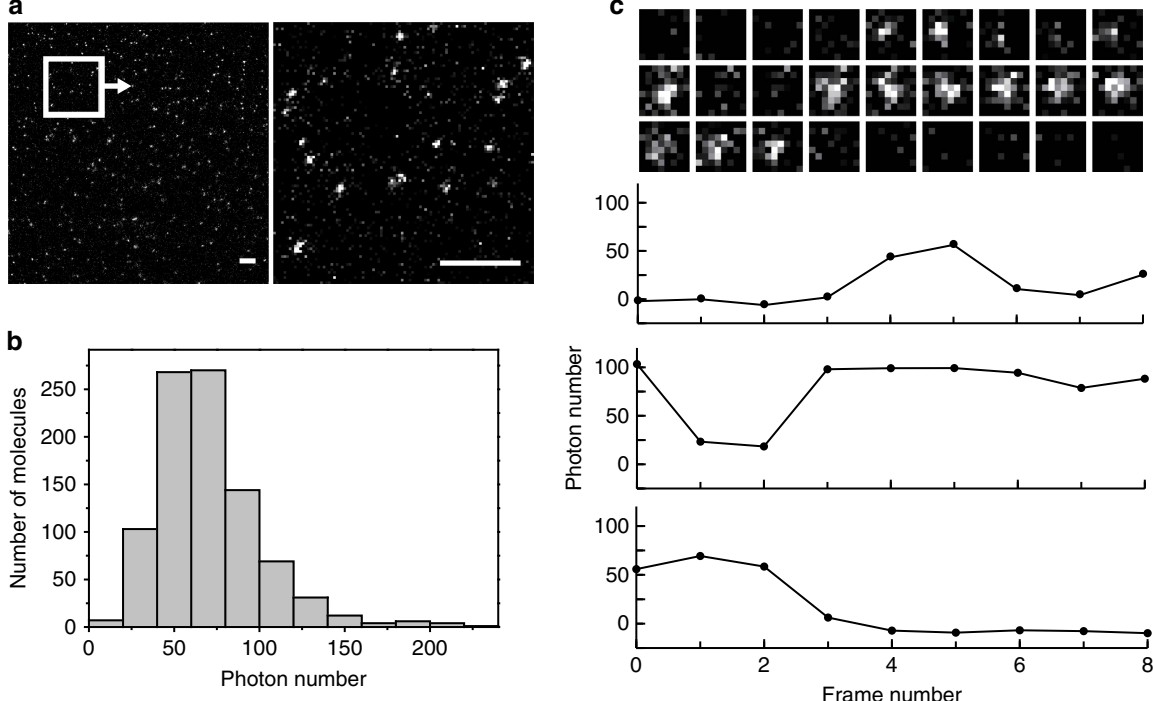

**Figure 2 | Detection of luminescence from single GeNL molecule. (a)** Luminescence images of single GeNL molecule (left). The exposure time was 180 s. The magnified image of the square inset (right). Scale bars, 10 μm. **(b)** A histogram of the average photon numbers emitted from single spots (*N* = 919). **(c)** A time series of single GeNL molecule (above). Time course in the total photon numbers within the ROIs (below).

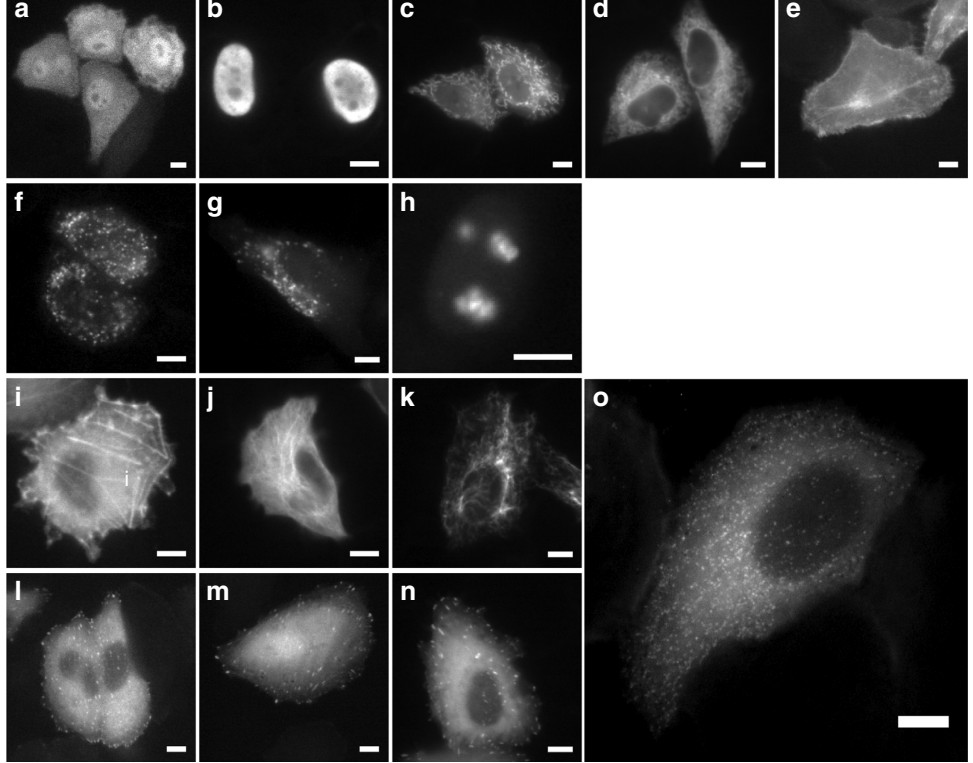

**Figure 3 | Luminescence imaging of HeLa cells expressing eNLs targeted to the various cellular compartments.** HeLa cells expressing GeNL in cytosol (**a**), nucleus (**b**), mitochondria (**c**), ER (**d**), inner plasma membrane (**e**), peroxisome (**f**), lysosome (**g**), nucleoli (**h**), actin (**i**), microtubule (**j**), vimentin (**k**), vinculin (**l**), zyxin (**m**), paxillin (**n**) and CeNL in a clathrin-coated pit (**o**). Scale bars, 10 μm.

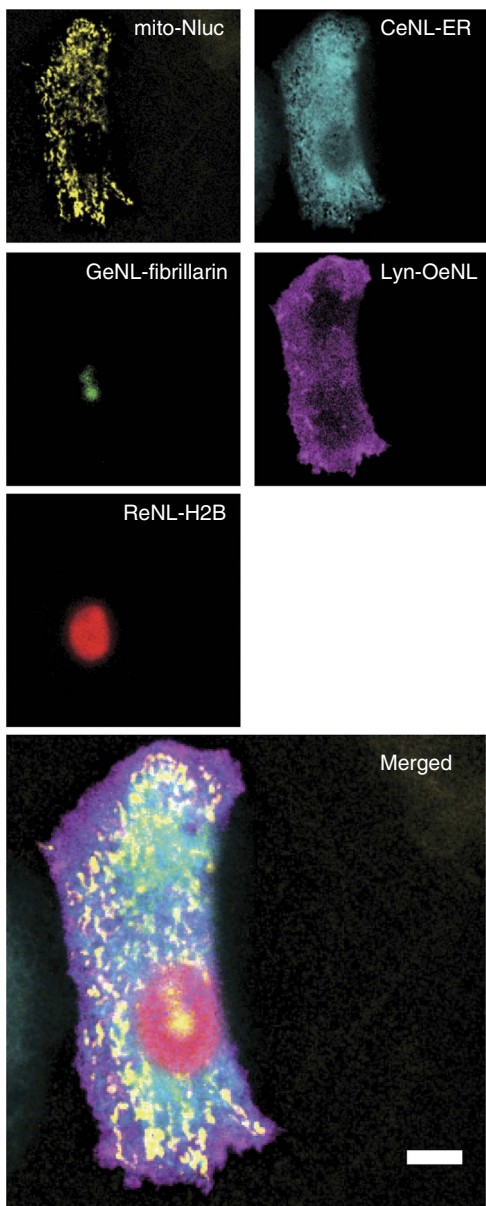

**Figure 4 | Multicolour luminescence image of subcellular structures.**
Mitochondria (mito-Nluc); ER (CeNL-ER); nucleoli (GeNL-fibrillarin); plasma membrane (Lyn-OeNL); and nucleus (ReNL-H2B). Each luminescence signal was separated by linear unmixing. Scale bar, 10 μm.

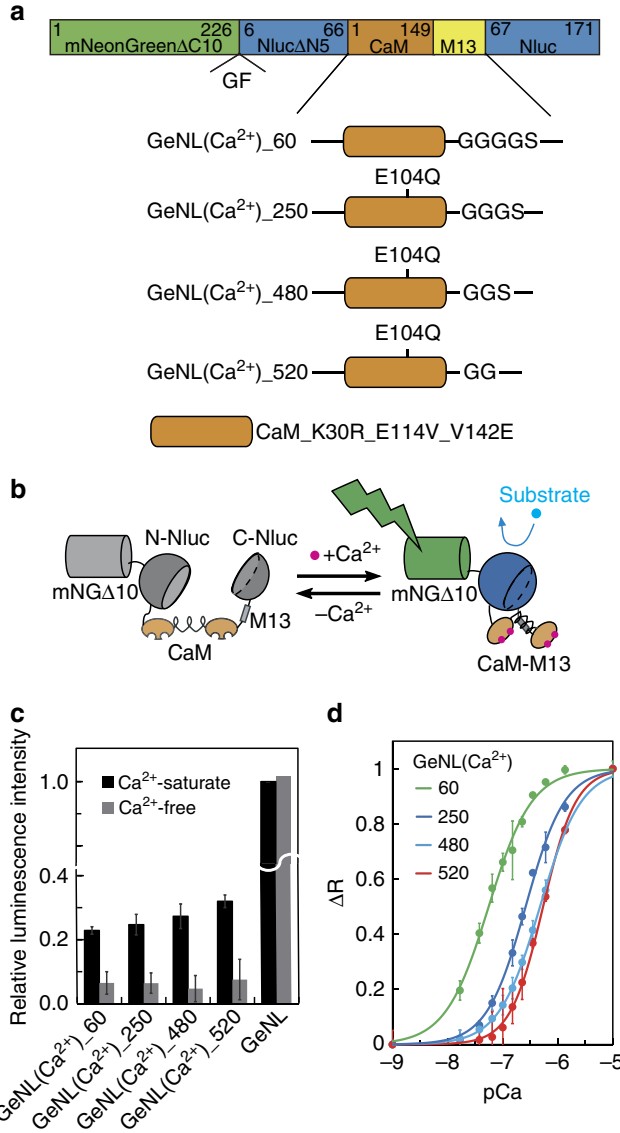

**Figure 5 | Luminescent Ca$^{2+}$ indicators based on GeNL.** (**a**) Domain structure of four affinity variants of GeNL(Ca$^{2+}$). (**b**) Schematic explanation of Ca$^{2+}$-sensing mechanism in GeNL(Ca$^{2+}$). (**c**) Relative brightness of recombinant GeNL, GeNL(Ca$^{2+}$) affinity variants, with or without Ca$^{2+}$. (**d**) Ca$^{2+}$ titration curves for four affinity variants of GeNL(Ca$^{2+}$). Measurements in **c,d** were performed at least in triplicate, and the averaged data and s.d.'s are shown.

Related Gene (hERG) inhibition, we detected irregular beating, like an arrhythmia[26] (Fig. 7b). These results indicate that our luminescence-based Ca$^{2+}$ indicator can be used not only for imaging fast physiological events but also for analysing drug effects on cardiomyocyte function.

## Discussion

Through systematic protein engineering, we identified five colour variants of bright LPs across the visible spectrum. The use of eNLs allowed us to perform five-colour luminescence imaging of subcellular structures. A remarkable feature of eNL is signal intensity sufficient to visualize luminescence from single molecules *in vitro* and single clathrin-coated pit in living cells. We also developed an eNL-based Ca$^{2+}$ indicator with a 500% signal change. This indicator enabled long-term and fast Ca$^{2+}$ imaging in cardiomyocytes.

Luminescence imaging in general has other limitations that the eNLs do not overcome. First, luminescence signal decays over time by consumption of the luminescent substrate. This issue may be possible to overcome by continuous perfusion with fresh luminescent substrate. Second, the luminescent substrates may affect cell behaviour. Coelenterazine is reported to possess high antioxidant activity against reactive oxygen species[27]. Since furimazine is an analogue of coelenterazine, it might perturb cellular physiology by disruption of signal cascades involving reactive oxygen species. To minimize the potential for this effect most of our experiments use <20 μM furimazine, which does not affect cell viability and morphology[4]. Third, coelenterazine and its analogues are reported to be a substrate for multidrug resistance (MDR1) P-glycoprotein transports[28]. However, as we could easily detect luminescence signals of eNLs in various cellular

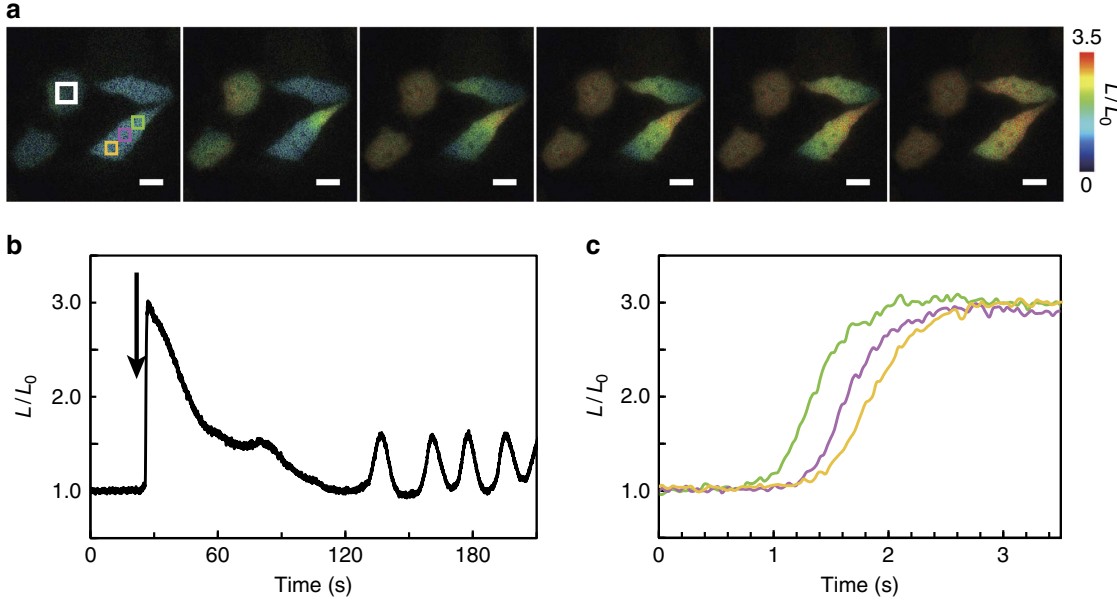

**Figure 6 | High-speed Ca$^{2+}$ imaging with GeNL(Ca$^{2+}$)_480 in HeLa cells.** (**a**) A series of intensity modulated display pseudo-coloured ratio images showing Ca$^{2+}$ concentration change by histamine stimulation. Scale bars, 20 μm. (**b**) Long-range time course of the $L/L_0$ ($L$: luminescence intensity at arbitrary time; $L_0$: luminescence intensity before stimulation). Arrow indicates the time point of 10 μM histamine addition in the ROI (white rectangle). (**c**) Short-range time courses of the $L/L_0$ ratio change in ROIs (green, magenta and yellow boxes in **a**).

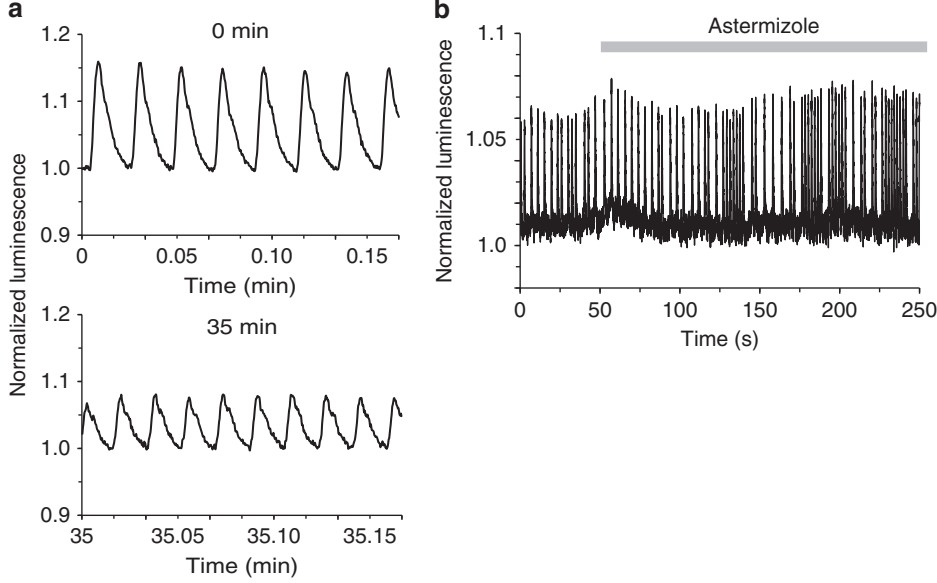

**Figure 7 | High-speed Ca$^{2+}$ imaging with GeNL(Ca$^{2+}$) in hiPSC-CMs.** (**a**) Time course of the luminescence signal of GeNL(Ca$^{2+}$)_520 at around 0 and 35 min. (**b**) Time course of luminescence signal of GeNL(Ca$^{2+}$)_520 before and after treatment with 1 μM astemizole.

compartments, furimazine membrane permeability within the cells does not appear limiting.

The design of GeNL-based functional indicators could be expanded to other colour variants and biomolecules such as cyclic AMP and ATP[9].

Furthermore, we believe that eNLs are promising reporters for endogenous proteins. It is now feasible to create endogenously tagged proteins using CRISPR/Cas9-mediated genome editing. The enhanced luminescence signal of eNL allows detection of low-copy-number proteins, which is difficult by fluorescence imaging, especially where the cellular substrate has autofluorescence.

The red-shifted emission of ReNL is also important for *in vivo* imaging because it can be distinguished from the background signal caused by self-oxidation of substrate in tissue. It can also overcome the optical scattering of luminescence signal that occurs in deep tissue[29]. Thus, we expect that eNLs could facilitate detection of rare, but significant cell populations (cancer stem cells and so on) in living mice in addition to functional imaging in freely moving animals.

## Methods

**General.** DNA oligonucleotides used for cloning and construction of gene libraries were purchased from Hokkaido System Science. The sequences of all the

oligonucleotides used in this study are provided in Supplementary Table 3. KOD-Plus (Toyobo Life Science) was used for non-mutagenesis PCR amplification. Products of PCR and restriction enzyme (RE) digestion were routinely purified using preparative agarose gel electrophoresis followed by DNA isolation using a QIAEX II gel extraction kit (Qiagen). Restriction endonucleases were purchased from New England Biolabs or Takara-bio, and used according to the manufacturer's recommended protocol. Ligations were performed using T4 ligase in Rapid Ligation Buffer (Promega). Small-scale plasmid DNA preparation was performed by alkaline lysis of the bacterial pellet obtained from 1.5 ml of LB liquid culture followed by ethanol precipitation of the DNA. Large-scale plasmid DNA preparations were performed by alkaline lysis of the bacterial pellet derived from 200 ml of Luria-Bertani (LB) liquid culture followed by isopropanol precipitation, PEG 8000 precipitation and two rounds of phenol/chloroform extraction. cDNA sequences for all constructs were read by dye terminator cycle sequencing using the BigDye Terminator v1.1 Cycle Sequencing kit (Life Technologies). The luminescent substrate coelenterazine-h was purchased from Wako Chemicals, and furimazine was synthesized according to a previous report[4]. Cultured cells were not tested for the presence of *Mycoplasma*, as such contamination would not affect the conclusions made on the basis of our results.

**Modelling.** The primary amino-acid sequence of Nluc[4] was used as input to the I-TASSER structure prediction server[30] using default parameters. The generation of figures depicting structures was done using UCSF chimera[31].

**Construction of five colour eNL variants.** cDNA of mNeonGreen was provided by Allele Biotechnology. The deletion mutant libraries of mNeonGreen–Nluc and Venus–Nluc fusion constructs were generated as follows[32]. The cDNAs of C-terminally deleted mNeonGreen mutants (mNGΔC0-10) and Venus mutants (VenusΔC0-12) were amplified by PCR. They were digested with BamHI and KpnI. The cDNAs of N-terminally deleted Nluc mutants (NlucΔN1-5 for mNG and NlucΔN1-6 for Venus) were amplified by PCR and digested with EcoRI and KpnI. The digested PCR fragments were gel-purified, mixed together and cloned in-frame into the BamHI/EcoRI sites of pRSET$_B$ (Invitrogen) for bacterial expression (Supplementary Fig. 1). After screening, the linker amino acids (encoded by KpnI, –Gly–Thr–) of deletion mutants was randomized by inverse PCR techniques (nucleotide sequence NNKNNK, where $N = A$, G, C and T; and $K = G$ and T) to generate 400 amino-acid combinations (1,024 nucleotide combinations; Supplementary Fig. 3).

Considering the 230th–239th residues of mTQ2 and 464th–476th residues of tdTomato could be removed without disrupting the function and the ΔN4Nluc was selected in the screening of GeNL, we chose mTQ2ΔC10-GT-ΔN4Nluc and tdTomatoΔC8-GT-ΔN4Nluc as templates for CeNL and ReNL development. We constructed linker libraries by systematic truncation of the connecting region between either mTQ2 or tdTomato and Nluc using inverse PCR techniques (Supplementary Fig. 5). For each library, two randomized amino-acid residues (nucleotide sequence NNKNNK, where $N = A$, G, C and T; and $K = G$ and T) were placed between the C terminus of either mTQ2 or tdTomato and the N terminus of Nluc, to generate 400 amino-acid combinations (1,024 nucleotide combinations).

The cDNAs of mKOκ with various length flexible linkers (no linker to GGSGGSGGS for 5′-end of mKOκ and no linker to TLGMDELYK for 3′-end) were amplified by PCR. They were digested with XhoI and SacI, then gel-purified and mixed together. XhoI and SacI restriction sites were also introduced at the 50th/51st insertion site of the Nluc moiety in pRSET$_B$_Nluc. This fragment was then ligated with the mKOκ fragment mixture to yield the pRSET$_B$_mKOκ@51_Nluc library (Supplementary Fig. 7c).

The nucleotide sequences of GeNL, YeNL, CeNL, ReNL and OeNL are in Supplementary Note 4.

**Covalent labelling of Nluc with eosin maleimide dye.** Eosin-5-maleimide was purchased from Molecular Probes (no. E-118). To achieve the labelling at specific positions, we substituted the endogenous cysteine of Nluc (166th residue) with alanine, and four glycine residues (50th, 66th, 97th and 136th) were substituted with cysteine, respectively. The mutated Nluc was expressed in *Escherichia coli* and purified as explained below. The protein (50 μM) was pre-incubated with 5 mM dithiothreitol in 20 mM HEPES buffer (pH = 7.4) for 30 min at room temperature. The solutions were then changed to fresh 20 mM HEPES buffer (pH = 7.4) using a NAP-5 column (GE-Healthcare). The proteins were incubated with 100 μM eosin-5-maleimide in 20 mM HEPES buffer (pH = 7.4) for 1 h at room temperature. The solutions were changed to fresh 20 mM HEPES buffer (pH = 7.4) using a NAP-5 column to remove the free eosin-5-maleimide. The luminescent spectral measurement was performed as described in 'LP characterization' below.

**Screening of mutants with bright and high-FRET efficiency.** After transforming *E. coli* JM109 (DE3) with the mutagenized DNAs, we spread the bacterial cells homogeneously over 95 mm agar plates and incubated them at 37 °C for 12 h, before sitting for 2 days at room temperature to allow the chromophore maturation. Bright mutants with high-FRET efficiency were screened and selected in two steps. Initially, the *E. coli* colonies were poured with a phosphate-buffered saline (PBS) solution supplemented with 5 μM coelenterazine-h and examined with

an LAS-1000 luminescence imaging system (GE Healthcare). Bright colonies were picked, and those mutants were inoculated in the 96-well plate supplemented with 100 μl liquid LB medium. The 96-well plate was incubated at 23 °C for 3 days before luminescence spectra were measured using a micro-plate reader (SH-9000; Corona Electric). A final concentration of 5 μM coelenterazine-h was used as the luminescent substrate for this measurement. Luminescent spectra were normalized at the 450 nm luminescence intensity and mutants with a high-FRET ratio (ratio of peak intensity at 520/450 nm for mNeonGreen, 530/450 nm for Venus, 565/450 nm for mKOκ and 585/450 nm for tdTomato) picked and subjected to DNA sequencing and protein characterization. Because the emission peaks of Nluc (donor, ~460 nm) and mTQ2 (acceptor, 480 nm) were too close to discern, the cyan variants were directly purified and screened *in vitro* on the basis of brightness and FRET efficiency.

**Construction of mammalian expression vectors.** To ensure the robust expression of GeNL in mammalian cells, we replaced the wild-type codon with synthesized cDNA encoding the mNeonGreen with mammalian favourable codons obtained from Life Technologies (GeneArt Strings DNA Fragments). PCR-amplified eNLs were inserted into a pcDNA3 mammalian expression vector using BamHI and EcoRI RE sites. We localized eNLs to mitochondria, plasma membrane and nucleus, respectively, by replacing the Nano-lantern sequence with the eNL sequence in pcDNA3-CoxVIIIx2-Nano-lantern (a duplicated mitochondrial localization sequence derived from the subunit-VIII precursor of human cytochrome *c* oxidase (Cox-VIII) at the N terminus); pcDNA3-Nano-lantern-H2B (a DNA-binding protein histone 2B (H2B) at C terminus); and pcDNA3-lyn-Nano-lantern (a myristoylation and palmitoylation sequence from lyn kinase at N terminus)[9]. For peroxisome localization, PCR-amplified eNLs with SKL (a peroxisome localization sequence) were inserted into a pcDNA3 mammalian expression vector using BamHI and EcoRI RE sites. For nucleolus and ER localization, we replaced Phamret with eNLs in pcDNA3-Phamret-fibrillarin or pcDNA3-Phamret-ER (signal peptide from calreticulin at the N terminus and an ER retention signal at the C terminus) using BamHI and EcoRI or BamHI and KpnI RE sites[33]. A 5-amino-acid linker (GGSGGT) was inserted between the eNLs and ER retention signal sequences. For paxillin, zyxin and vimentin localization, we replaced the Kohinoor sequence with the GeNL sequence in pcDNA3-paxillin-Kohinoor, pcDNA3-zyxin-Kohinoor or pcDNA3-vimentin-Kohinoor using BamHI and EcoRI RE sites[34]. For β-actin localization, we replaced the Kohinoor sequence with the eNLs sequence in the pcDNA3-Kohinoor-β-actin vector using HindIII and KpnI RE sites. The open reading frames (ORFs) of vimentin, paxillin and zyxin included a 17-amino-acid linker peptide (GTGSGGGGSGGGGSGGS), and β-actin included a 20-amino-acid linker peptide (GGSGGSGGSGGSGGSGGEFQ IST). For clathrin localization, we replaced the Kohinoor sequence with the eNLs sequence in pEGFP-N1-Kohinoor-clathrin using NheI and BglII RE sites. Clathrin had a 12-amino-acid linker peptide (RSRAQASNSAVD). For β-tubulin localization of GeNL, we replaced the Kohinoor sequence with the GeNL sequence in pEGFP-N1-β-tubulin-Kohinoor using SalI and NotI RE sites. A 21-amino-acid linker (QSTGSGGGGSGGSTVPRARDP) was inserted between the GeNL and β-tubulin sequences. For β-tubulin localization of CeNL, YeNL, OeNL and ReNL, a β-tubulin sequence was inserted upstream of CeNL, YeNL, OeNL and ReNL in pcDNA3-CeNL, pcDNA3-YeNL, pcDNA3-OeNL and pcDNA3-ReNL using HindIII and EcoRI RE sites. The ORFs of β-tubulin included a 23-amino-acid linker peptide (QSTVPRARDPGSGGGSGGGSGEF). For vinculin localization, a vinculin sequence was inserted downstream of GeNL in pcDNA3-GeNL using KpnI and EcoRI RE sites. For lysosome localization, a LAMP sequence was inserted upstream of eNLs in pcDNA3-eNLs using HindIII and BamHI RE sites. The ORFs of vinculin and LAMP included a 16-amino-acid linker peptide (GTGGGGSGG GGSGGSG) and a 17-amino-acid linker peptide (GTGSGGGGSGGGGSGGGS). The cDNAs of LAMP and vinculin were amplified from the plasmids, mRuby2-Lysosomes-20 and pEGFP Vinculin, which were gifts from Michael Davidson and Kenneth Yamada (Addgene plasmid # 55902 and 50513, respectively). See Supplementary Table 3 for the polynucleotides used in this study.

**Protein expression and purification.** LP with an N-terminal polyhistidine tag was expressed in *E. coli* (JM109 (DE3)) at 23 °C for 65 h in LB bacterial growth medium supplemented with 0.1 mg ml$^{-1}$ carbenicillin. Cells were collected and ruptured with a French press (Thermo Fisher Scientific), and recombinant proteins were purified from the supernatant using Ni-NTA agarose affinity columns (Qiagen) followed by buffer-exchange (20 mM HEPES, pH 7.4) gel filtration (PD-10 column, GE Healthcare). The whole purification process after rupture was conducted on ice to avoid protein degradation. The protein concentration was determined by Bradford method (Protein assay kit, Bio-Rad).

**LP characterization.** Recombinant proteins and furimazine were diluted with 20 mM HEPES buffer (pH 7.4), and emission spectra were measured with a photonic multichannel analyser PMA-12 (Hamamatsu Photonics) at room temperature using 500 ms exposures. The final concentrations of proteins and substrate were 100 nM and 25 μM, respectively. Furimazine was used for eNLs and Nluc as a substrate; coelenterazine-h was used for the other proteins. Experiments

were performed at least in triplicate, and the averaged data were used for further analysis.

**Luminescent quantum yield and kinetic parameters.** The luminescent quantum yields were estimated from the total light output by the complete consumption of 0.05 pm of furimazine. The luminescent quantum yields were measured in triplicate with a micro-plate reader (SH-9000, Corona Electric). The photon sensitivity of the detector was calibrated with luminol chemiluminescence under following reaction mixture: 200 nM horseradish peroxidase; 200 nM luminol; and 2 mM hydrogen peroxide in $K_2CO_3$ aqueous solution[35]. The concentration of luminol (Wako, Osaka, Japan) was characterized by the absorbance at 347 nm with using the extinction coefficient of $7,640 \, M^{-1} \, cm^{-1}$ as reported previously[35]. The wavelength characteristics of the detector were adjusted in relation to the photonic multichannel analyzer PMA-12 (Hamamatsu Photonics). The final concentrations of LP and furimazine were 1 nM and 500 pM, respectively. The protein and substrate solutions were diluted with HEPES buffer (50 mM HEPES, pH 7.5) supplemented with $<0.1\%$ casein.

Kinetic parameters were measured from the reactions of final 10 pM LP with final furimazine concentrations of 0.025, 0.051, 0.10, 0.20, 0.41, 0.81, 1.6, 3.3, 6.5 and 13 μM, respectively. The initial reaction velocities were measured as the integrated luminescence intensities for the initial 12 s. Michaelis–Menten constants $(K_m)$ and maximum reaction velocities $(V_{max})$ were estimated from the nonlinear fitting to the Michaelis–Menten equation using Origin7 software (OriginLab).

**Detection of luminescence from single-molecule GeNL.** To immobilize the GeNL protein on the glass surface, we prepared the Ni-NTA agarose glass cover-slips following a previous study[36]. The fluorescent beads were used for focusing on the glass surface. First, fluorescent beads (FluoSpheres sulfate microspheres, 0.2 μm diameter, yellow-green fluorescent, no. F8848, Invitrogen) in MOPS/KCl buffer (10 mM MOPS and 100 mM KCl, pH = 7.2) were placed on the Ni-NTA agarose glass and incubated for 5 min and removed. Subsequently, 10 pmol of GeNL purified protein in MOPS/KCl buffer was placed on it and incubated for 5 min and washed with MOPS/KCl buffer three times to remove the unbound proteins. Just before observation, 50 μM furimazine was added. To detect the luminescence from a single molecule, an inverted microscope (LV-200, Olympus) equipped with a × 100 objective (Olympus, UPlanSApo, numerical aperture 1.4) and × 0.5 relay lens was used. Emission signals were detected by a cooled EM-CCD (electron-multiplying, charge-coupled device) camera (ImagEM, Hamamatsu Photonics) with 1 × 1 binning settings and 180 s exposure time.

A series of images was processed by Fiji. The background (defined as mean intensity at a region where no luminescence spot was detected) was subtracted using ImageJ's built-in function. Images were transformed to a 32-bit float and the count of each pixel was converted to photon number (designated as raw images; see also Supplementary Note 2). A median filter (radius size = 1) was used to smooth the images. Then luminescence spots were segmented by means of the Particle Track Analysis (PTA ver1.2) plug-in of ImageJ (https://github.com/arayoshipta/projectPTAj) to define square ROIs (filtered by minimum intensity 0.9, size $>5$ pixel$^2$). The total photon number at each ROI was calculated from the raw images and plotted in a histogram (Fig. 2b) using Origin7 software.

**Preparation and luminescence imaging of HeLa cell.** HeLa (RIKEN BRC) cells were cultured on collagen-coated 35 mm glass-bottom dishes in Dulbecco's modified Eagle's medium (DMEM) supplemented with 10% fetal bovine serum (FBS). The next day, HeLa cells ($\sim$70% confluency) were transfected with 4.0 μg plasmid DNA using Lipofectamine 2000 Transfection Reagent (Life Technologies) according to the manufacture's recommended protocol. Medium was replaced after 8 h, and the cells grown for an additional 16 h in a $CO_2$ incubator (Sanyo) at 37 °C in 5% $CO_2$. HeLa cells were washed with phenol red-free DMEM/F12 and imaged in phenol red-free DMEM/F12. Just before observation, 20 μM furimazine was added to the imaging medium. To observe eNL signals in living HeLa cells, an inverted microscope (LV-200, Olympus) equipped with a × 100 objective (Olympus, UPlanSApo, numerical aperture 1.4) and × 0.5 relay lens was used. Emission signals were detected by an EM-CCD camera (ImagEM, Hamamatsu Photonics) with 1 × 1 (for eNL) or 2 × 2 (for GeNL($Ca^{2+}$)) binning settings. To observe the localization of clathrin light chain labelled with CeNL in living HeLa cells, an inverted microscope (IX83, Olympus, Japan) equipped with a × 100 objective (Olympus, UPlanSApo, numerical aperture 1.4) was used. Emission signals were detected by an EM-CCD camera (Evolve 512, Photometrics) with 1 × 1 binning.

**Preparation and $Ca^{2+}$ imaging of GH3 cells.** GH3 (rat pituitary tumour, ATCC CCL-82.1) cells were cultured on poly-D-lysine-coated 35 mm glass-bottom dishes in DMEM/F12 supplemented with 15% horse serum and 2.5% FBS. For $Ca^{2+}$ imaging using either GCaMP3 or GeNL($Ca^{2+}$), GH3 cells ($\sim$70% confluency) were transfected with 4.0 μg plasmid DNA using Lipofectamine 2000 Transfection Reagent (Life Technologies) according to the manufacture's recommended protocol. The medium was replaced after 8 h, and the cells were grown for an additional 16 h in a $CO_2$ incubator (Sanyo) at 37 °C in 5% $CO_2$. For $Ca^{2+}$ imaging using Fura-2, 5.0 μM Fura-2-AM (Dojindo Laboratories, F016) was loaded onto

GH3 cells in Hanks' Balanced Salt solution (Sigma) supplemented with 1 × PowerLoad (Invitrogen) for 1 h at 37 °C. The GH3 cells were washed with phenol red-free DMEM/F12 and imaged in phenol red-free DMEM/F12. Just before observation of GeNL($Ca^{2+}$), 20 μM furimazine was added to the imaging medium.

Images were captured using an inverted microscope (IX83, Olympus, Japan) equipped with a × 40 objective (Olympus, UApo/340, numerical aperture 1.35) or a × 60 objective (Olympus, PlanApo, numerical aperture 1.4), EM-CCD camera (Evolve 512, Photometrics). The dual-excitation ratio imaging of Fura-2 was conducted with Semrock FF01-340/26 for shorter excitation wavelengths, FF01-387/11 for longer excitation wavelengths, an FF409-DiO2 dichroic mirror and an FF01-510/84 emission filter. The excitation filters for dual excitation images of Fura-2 were alternated using a filter exchanger with 100 ms exposure times for each channels, while sample were illuminated only during exposure time. Images of Fura-2 or GeNL($Ca^{2+}$) were acquired at 1.3 Hz frame rate with 2 × 2 binning. The $Ca^{2+}$ imaging of GCaMP3 was conducted with Omega 475AF20 excitation filter, Semrock FF495-Di03 dichroic mirror and an FF01-525/45 emission filter. The images of GCaMP or GeNL($Ca^{2+}$) were acquired at 33 Hz frame rate with 2 × 2 binning and 30 ms exposure time. Cells were maintained on the microscope at 37 °C by using a stage-top incubator (Tokai Hit, Fujinomiya, Japan).

**Multicolour imaging with linear unmixing.** For multicolour luminescence imaging, five images were acquired with Semrock FF01-447/60 filter (for Nluc), Olympus BA460-510CFP (for CeNL), Semrock FF01-525/35 filter (for GeNL), Semrock FF01-562/40 (for OeNL) and Semrock FF01-593/40 (for ReNL). Five HeLa cells (expressing each fusion construct) were imaged with identical conditions to determine the coefficients for linear unmixing. The signals from NLuc, CeNL, GeNL, OeNL and ReNL were then separated by linear unmixing using these coefficients by PrizMage software (Molecular Devices).

**Searching for insertion site of NanoLuc.** The Mutation Generation System Kit (Thermo Fisher Scientific) was used to construct the insertion mutation library of Nluc variants. Following the manufacturer's protocols, an *in vitro* transposition reaction was performed with 280 ng of pRSET$_B$-Nluc and the M1-Kan$^R$ Entranceposon that contains the kanamycin resistance gene. The reaction mixture was used to transform *E. coli* XL10-Gold. The transformation reaction was plated on two LB/agar/amp plates supplemented with kanamycin (25 μg ml$^{-1}$) and incubated for 12 h at 37 °C. Bacterial colonies were scraped from all the plates and then plasmid DNA was purified. The plasmid DNA was digested with BamHI and EcoRI and the resulting fragments were separated by agarose gel electrophoresis. The 1.6 kb (0.5 kb Nluc gene + Entranceposon) band was excised, isolated and ligated with appropriately digested pRSET$_B$. The ligation reaction was used to transform *E. coli* as described above. The transformation reaction was diluted into LB/amp and incubated overnight at 37 °C with shaking at 225 r.p.m. Plasmid DNA was isolated from this culture, digested with NotI to remove the Kan$^R$ gene, and the resulting fragments were separated by agarose gel electrophoresis. The 3.4 kb fragment (modified Nluc gene + pRSET$_B$) was excised and the DNA isolated as described above. The isolated DNA fragment was self-ligated to circularize the plasmid and thus provide the library of Nluc variants with random 15 bp (five codon) insertions.

The plasmid library of Nluc with random insertions was used to transform the chemically competent *E. coli* strain JM109(DE3). The transformed bacteria were plated on LB agar. Plates were incubated for 12 h at 37 °C. Following a 20 h incubation at 23 °C, a PBS solution containing 5 μM coelenterazine-h was added to the *E. coli* colonies, and were examined using an LAS-1000 luminescence imaging system (GE Healthcare). The 30 brightest colonies were picked up and cultured in liquid LB medium for 12 h. They were subjected to plasmid DNA isolation and subsequent DNA sequencing as described above.

To determine the insertion sites of Nluc that yield a large signal increase with $Ca^{2+}$, we used the TorA protein export plasmid (pTorPE) in which a $Ca^{2+}$ indicator localized to the *E. coli* periplasm can be placed on $Ca^{2+}$-bound states and easily extracted by cold osmotic shock[37]. We subcloned Nluc into pTorPE using SalI and HindIII RE sites. To obtain cDNA fragment encoding the CaM–M13 peptide of Nano-lantern($Ca^{2+}$), we digested a pRSET$_B$_Nano-lantern($Ca^{2+}$)_600 with NcoI and SacI. These restriction sites were also introduced at the identified insertion sites of pTorPE_Nluc (Supplementary Fig. 15). These fragments were then ligated to yield pTorPE_CaM-M13_Nluc and then mixed together. DH10B bacterial cells were transformed and spread on LB/amp agar plates supplemented with L-arabinose (0.02%). Following a 12 h incubation at 37 °C, a PBS solution containing 5 μM coelenterazine-h was added to the *E. coli* colonies, before examination using an LAS-1000 luminescence imaging system (GE Healthcare). The brightest colonies were picked and cultured in liquid LB medium for 12 h. The brightest colonies were subjected to further DNA purification and sequencing. An aliquot of *E. coli* suspensions was also used for secondary screening, in which $Ca^{2+}$-dependent change in luminescence was assessed using the protein extracted from periplasmic fraction of *E. coli*. Extraction of periplasmic protein from *E. coli* was performed by cold osmotic shock procedure as described before[37]. Briefly, bacterial cells were collected by centrifugation at 13,000g for 2 min at 4 °C and gently resuspended in 500 μl of ice-cold pH 8.0 buffer containing 30 mM Tris-HCl, 1 mM EDTA and 20% sucrose. After 5 min of gentle agitation on ice, the bacteria were again pelleted by centrifugation (9,000g for 5 min at 4 °C), resuspended in

400 μl of ice-cold 5 mM $MgSO_4$ and gently agitated for 10 min on ice. Following centrifugation to pellet the intact bacteria (9,000g for 5 min at 4 °C), the supernatant (the osmotic shock fluid containing the periplasmic protein fraction) was collected. The $Ca^{2+}$-dependent change in luminescence was measured by a micro-plate reader as described in 'In vitro characterization of GeNL($Ca^{2+}$)'.

**Construction of GeNL($Ca^{2+}$).** NcoI and SacI restriction sites were introduced at the 66/67 and 69/70 insertion sites of the Nluc moiety in $pRSET_B$_GeNL. This fragment was then ligated with a CaM–M13 fragment to yield $pRSET_B$_CaM-M13@67_GeNL and $pRSET_B$_CaM-M13@70_GeNL. To restore the brightness of $pRSET_B$_CaM-M13@67_GeNL, we performed error-prone PCR for CaM–M13 and a C-terminal fragment of Nluc under conditions to achieve a mutation rate of 5 bp kb$^{-1}$. Primers containing NcoI/EcoRI were used for re-insertion of CaM–M13_Nluc (67th–171th) mutant cDNA into $pRSET_B$_CaM-M13@67_GeNL. JM109(DE3) bacterial cells were transformed and then spread on LB/amp agar plates. Following a 12 h incubation at 37 °C, a PBS solution containing 5 μM coelenterazine-h was added to the E. coli colonies before examination using a LAS-1000 luminescence imaging system (GE Healthcare). The brightest colonies were picked and cultured in liquid LB medium for 12 h. They were subjected to further DNA purification and sequencing. The mutant with the largest brightness and signal increase was designated GeNL($Ca^{2+}$)_480 (latter number indicates $K_d$ value to $Ca^{2+}$).

To generate affinity variants of GeNL($Ca^{2+}$), we introduced various length linkers between CaM and M13 by Quik Change site-directed random mutagenesis[38]. For mammalian expression experiments, GeNL($Ca^{2+}$) cDNA digested with BamHI and EcoRI was cloned into the BamHI/EcoRI site of the pcDNA3 vector. The nucleotide sequences of GeNL($Ca^{2+}$)_60, GeNL($Ca^{2+}$)_250, GeNL($Ca^{2+}$)_480 and GeNL($Ca^{2+}$)_520 are in Supplementary Note 4.

For the AAV expression system, pHelper and pAAV-DJ were obtained from Cell Biolabs, Inc[39]. The cDNA of GeNL($Ca^{2+}$)_520 was amplified by PCR using a sense primer containing a BamHI site followed by a Kozak sequence and a reverse primer containing an EcoRI site followed by a stop codon. The cDNA for GeNL($Ca^{2+}$)_520 replaced the ArchT-GFP sequence in pAAV-CAG-ArchT-GFP. pAAV-CAG-ArchT-GFP was a gift from Edward Boyden (Addgene plasmid #29777).

**In vitro characterization of GeNL($Ca^{2+}$).** Emission intensity of the purified proteins was measured using a micro-plate reader (SH-9000, Corona Electric). A final concentration of 5 μM coelenterazine-h was used as the luminescent substrate for these measurements. Experiments were performed at least in triplicate, and the averaged data were used for further analysis. $Ca^{2+}$ titrations were performed by the reciprocal dilution of $Ca^{2+}$-saturated and $Ca^{2+}$-free buffers containing 10 mM MOPS, 100 mM KCl and 10 mM EGTA with or without 10 mM $Ca^{2+}$ added as $CaCO_3$, at pH 7.2, 25 °C. The free $Ca^{2+}$ concentrations were calculated using 0.15 μM for the apparent $K_d$ value of EGTA for $Ca^{2+}$. The $Ca^{2+}$ titration curve was used to calculate the apparent $K_d$ value by nonlinear regression analysis. The averaged data were fitted to a single Hill equation using Origin7 software (OriginLab).

**AAV production and infection.** HEK293T (RIKEN BRC Cell Bank RCB2202) cells were grown in DMEM (Sigma) containing heat-inactivated 10% FBS at 37 °C in 5% $CO_2$. Equal amounts of pHelper, pAAV-DJ and pAAV_CAG_GeNL($Ca^{2+}$)_520 were transfected by $FuGENE_{HD}$ transfection reagent (Roche) following the manufacturer's protocol. Three days after transfection, the viruses were collected. The solution containing AAV was distributed into small aliquots and stored at −80 °C.

**hiPSC-CM culture and imaging.** Human iPS cells (hiPS, 201B7, RIKEN BRC) were cultured with a primate ES medium (Reprocell Inc.) and a 4 ng ml$^{-1}$ human basic fibroblast growth factor (Wako Pure Chemical Industries, Ltd.) in an incubator with 5% $CO_2$ at 37 °C on a mouse feeder cell layer (SNL, CBA-316, Cell Biolabs, Inc.). Embryoid bodies of hiPSC-CMs were prepared with the following procedure[40]. After carefully removing feeder cells with CTK solution (Reprocell Inc.), the one 60 mm dish of 80–90% confluent hiPS colonies were collected to the 15 ml centrifuge tube and centrifuge at 1,000 r.p.m. for 30 s, aspirated ES medium and change medium to 1 ml differentiation medium (RPMI + PVA), pipetting with 1 ml tip (WATSON, BIO LAB) for 20 times to obtain cell cluster size around 150 μm. The RPMI + PVA medium consist of RPMI Media 1640 (with L-glutamine), 4 mg ml$^{-1}$ polyvinyl alcohol (P8136 Sigma-Aldrich, St. Louis, MO), 400 μM 1-thioglycerol, 1% chemically defined lipid concentrate (Thermo Fisher Scientific, 11905-031), 10 μg ml$^{-1}$ recombinant human insulin (I9278, Sigma-Aldrich), 25 ng ml$^{-1}$ human BMP4, 5 ng ml$^{-1}$ human FGF2 (both from R&D systems) and 10 μM Y-27632 (Stemgent, Cambridge, MA). The suspended hiPS cells were transferred to 35 mm dish (Ultra Low Attachment Culture Dish, Corning) and place in incubator with 5% $O_2$ and $CO_2$ at 37 °C. Two days after incubation, the medium was replaced with the RPMI + FBS medium consisting of RPMI Media 1640, 20% FBS and 400 μM 1-thioglycerol, and the dishes were transferred to an incubator with 5% $CO_2$ at 37 °C. Four days after incubation, 8–10 embryoid bodies were transferred to gelatinized 0.1% coverslips in six-well plates. The medium was replaced with fresh RPMI + FBS medium every 3 days afterwards. The embryoid bodies were visually assessed for contraction on 9 days after incubation. Then they were treated with AAV for 4 days before observation.

The hiPSC-CM was washed with Tyrode solution (Sigma) and imaged in Tyrode solution. Just before observation, 40 μM furimazine was added to the Tyrode solution. An inverted microscope (LV-200, Olympus) equipped with a × 100 objective (Olympus, UPlanSApo, numerical aperture 1.4) and × 0.5 relay lens was used. Emission signals were detected by an EM-CCD camera (ImagEM, Hamamatsu Photonics) with 4 × 4 (for GeNL($Ca^{2+}$)) binning settings. During the entire imaging period, the temperature was kept at 28 °C by a stage-top incubator. The background drift was manually subtracted using Origin7 software (OriginLab).

**Data availability.** The data that support the findings of this study are available from the corresponding author on request. The nucleotide sequences of CeNL, GeNL, YeNL, OeNL, ReNL, GeNL($Ca^{2+}$)_60, GeNL($Ca^{2+}$)_250, GeNL($Ca^{2+}$)_480 and GeNL($Ca^{2+}$)_520 have been deposited to GenBank/EMBL/DDBJ database under the following entry IDs: LC128714; LC128715; LC128716; LC128717; LC128718; LC128719; LC128720; LC128721; and LC128722, respectively. The nucleotide sequences of all constructs are also in Supplementary Note 4.

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

## Acknowledgements

We thank George McNamara (The University of Texas) for reading an early draft of this manuscript and insightful discussion; Yasuteru Urano and Mako Kamiya (The University of Tokyo) for kind instruction on luminescent substrate synthesis; Hiromi Imamura (Kyoto University) kindly lent the Fura-2 filter sets; Akihiro Yamanaka (Nagoya University, RIEM) for providing us with the AAV purification method; Promega Corporation for providing us the cDNA of NanoLuc; and Allele Biotechnology for providing us the cDNA of mNeonGreen. Molecular graphics and analyses were performed with the UCSF Chimera package. Chimera is developed by the Resource for Biocomputing, Visualization, and Informatics at the University of California, San Francisco (supported by NIGMS P41-GM103311). This work was supported by a grant for 'JST-SENTAN', the MEXT Scientific Research on Innovative Areas, 'Spying minority in biological phenomena' (No. 3306) and JSPS Core-to-Core Program, A. Advanced Research Networks to T.N., and the Wellcome Trust (WT098519MA) to M.J.D.

## Author contributions

T.N. conceived and coordinated the project; K.S., T.K., H.S., G.B., M.J.D., Y.A., M.N. and T.N. designed the experiments; K.S. and T.K. constructed and characterized eNLs with the support of M.N. and Y.A.; K.S. performed imaging; H.S. constructed and characterized the eosin-conjugated Nluc; G.B. and M.J.D. established the iPS cell culture and performed the differentiation of iPS cells into cardiomyocytes; K.S., M.N., Y.A. and T.N. wrote the paper with contributions from all authors.
