## [Peer Review File · Nature Communications]

Reviewer #1 (Remarks to the Author):

A. Suzuki and colleagues report optimization of luciferase:fluorescent proteins pairs for multi-spectral imaging. They take advantage of a very bright luciferase, and show successful pairing with 5 fluorescent protein acceptors that are well suited for imaging in the same sample. Also, of interest, is an engineered calcium sensor.

B. These will be an important set of reagents for bioluminescence applications, particularly for in vivo imaging. Single molecule and calcium imaging show intriguing potential.

C. Data and methodology are appropriate.

D. Statistics are fine.

E. Conclusions, etc, are fine.

F. Minor concerns:

a) abstract states that "Ca²⁺ indicators could be used to image long-term Ca²⁺ dynamics in ... cardiomyocytes, which is not feasible with fluorescence imaging". In what way? It's not clear what is meant here because GCaMP imaging of cardiomyocytes is pretty robust. For as nice as these reagents are, it's not clear that they can report fast processes more robustly than fluorescence methods.

b) Line 104, regarding "super resolution applications", it's not obvious how these reagents could be adapted for a super-resolution, especially of the type cited. I would suggest elaborating more fully or revising this out.

c) legend, figure 2D, Which color is which sensor?

d) There is no discussion of the potential and unknown downsides to use of bioluminescence probes. Specifically, coelenterazine-like compounds are known for their antioxidant activity, which 'could' affect important biochemical pathways critical to cell function (see cardiomyocyte/XROS). The reader ought to be made aware of these potential issues, particularly because the use of these

reagent are new and the effects of things like furimazine on cell function have not been fully characterized.

e) I didn't note the presence of a section describing the cyan-colored reagent spectra, as was present for the other colors (e.g. sup fig 4). Such information would be useful because the cyan spectra is really close to the luciferase.

G & H are ok. for abstract concerns, see above (F-a)

Reviewer #2 (Remarks to the Author):

This is a well described development of a novel and highly original set of Nano-Lanterns that takes advantage of energy transfer from the bright bioluminescence protein NanoLuc to various hues of fluorescent proteins to enable multi-colour imaging without external excitation. I think these will be of wide interest to the community and open new avenues for experimentation with these unique indicators.

A substantial body of work has been presented which has been carefully undertaken and is of high quality. The simultaneous five-colour imaging and calcium indicator work is particularly impressive. The only element of the imaging that I didn't find compelling was the claim that a single clathrin-coated pit was identified in Figure 1e. The resolution is not good enough and the location seems to be predominantly perinuclear. The figure legend to Figure 1f should also make clear that the outputs were submitted to spectral unmixing to separate out the colours.

The single molecule imaging (Figure 1d) simply reflects imaging of individual particles that could represent aggregate complexes containing the GeNL lantern. How convinced are the authors that this represents single molecules of GeNL rather than single aggregates? If they are single molecules, does each detected particle have the same brightness?

The manuscript is clearly written although doesn't reference all of the previous work with NanoLuc - which perhaps it should. Some discussion should be included regarding the requirements for the NanoLuc substrate and the need for it to equally access the Nano-lanterns in different cellular locations. It would be worth expanding Supplementary Table 2 to ensure that all of the Nano-lanterns have the same K_m for furimazine.

Reviewer #3 (Remarks to the Author):

This paper describes the development and experimental validation of five luminescent reporter constructs. The manuscript is well written and accessible, and the experimental methodology appropriate and rigorous. Nevertheless, there are some concerns regarding the utility and capabilities of these probes.

There is no doubt that these probes could be a useful addition to the imaging toolbox, however, the authors fail to demonstrate that they offer significant advantages over conventional fluorescent probes, as suggested (lines 36-39). There is no direct comparison between the new probes and conventional probes. Furthermore, the statement that long-term Ca²⁺ dynamics in iPS-derived cardiomyocytes is not feasible with fluorescent imaging (line 30) is not justified. Since Ca²⁺ imaging has been done in iPS cardiomyocytes, the authors need to demonstrate the advantage of their approach over conventional techniques before making extraordinary claims.

The claim of single molecule resolution is intriguing (line 101-104), however, based on the evidence provided in Fig 1d, Supp Fig 10 and Note 2, this may be more speculative than the authors suggest. Given that single molecule imaging could be an important application for the probe then the claim should be better supported and included in the main body of the text, not as a supplementary note.

Reviewer #1

A. Suzuki and colleagues report optimization of luciferase:fluorescent proteins pairs for multi-spectral imaging. They take advantage of a very bright luciferase, and show successful pairing with 5 fluorescent protein acceptors that are well suited for imaging in the same sample. Also, of interest, is an engineered calcium sensor.

B. These will be an important set of reagents for bioluminescence applications, particularly for in vivo imaging. Single molecule and calcium imaging show intriguing potential.

C. Data and methodology are appropriate.

D. Statistics are fine.

E. Conclusions, etc, are fine.

We are very grateful for Reviewer1's comments on our work.

F. Minor concerns:

a) abstract states that "Ca²⁺ indicators could be used to image long-term Ca²⁺ dynamics in ... cardiomyocytes, which is not feasible with fluorescence imaging". In what way? It's not clear what is meant here because GCaMP imaging of cardiomyocytes is pretty robust. For as nice as these reagents are, it's not clear that they can report fast processes more robustly than fluorescence methods.

We agree that Ca²⁺ imaging of cardiomyocytes has been conducted by fluorescent probes such as GCaMP or Ca²⁺-sensitive organic dyes, eg Ref. 24. The comments of the reviewer prompted us to characterize the performance of our probe with the fluorescence toolbox of Ca²⁺ indicators GCaMP3 and Fura-2, in GH3 cells (rat pituitary tumor). GH3 cells exhibit repeated spontaneous Ca²⁺ spiking, similar to that seen in stem-cell derived or neonatal cardiomyocyte models.

Although all indicators showed a dynamic response, we found that the signal-to-noise ratio of GeNL(Ca²⁺)₄₈₀ (SNR 120±12 at 1.3 Hz frame rate, 74±3.0 at 33 Hz frame rate) was superior to that of Fura-2 (SNR 9.6±0.71 at 1.3 Hz frame rate), but inferior to that of GCaMP3 (SNR 590±25 at 33 Hz frame rate). We also found differences in the

duration of recording possible. Spontaneous Ca^{2+} spikes in GH3 cells expressing GeNL (Ca^{2+})₄₈₀ were visualized over 20 min. In contrast, the Fura-2 signal was much reduced 10 min into the observation due to phototoxic and photobleaching effects. These results suggest that GeNL(Ca^{2+}) reported the Ca^{2+} dynamics more robustly than Fura-2, but less than GCaMP3. Thus we have toned down our claim in the previous version by removing the sentence “*which is not feasible with fluorescence imaging*” and put the additional text and data in the results section and Supplementary Figure 16, respectively, as follow:

“We also made side-by-side comparisons of its performance with the benchmark Ca^{2+} indicators, GCaMP3 (Ref. 22) and Fura-2 (Ref. 23), in GH3 cells (rat pituitary tumor) which show spontaneous Ca^{2+} spikes (**Supplementary Fig. 16**). All indicators produced a dynamic signal trace. The signal to noise ratio of GeNL(Ca^{2+})₄₈₀ (SNR 120 ± 12 at 1.3 Hz frame rate, 74 ± 3.0 at 33 Hz frame rate) was superior to that of Fura-2 (SNR 9.6 ± 0.71 at 1.3 Hz frame rate), but inferior to that of GCaMP3 (SNR 590 ± 25 at 33 Hz frame rate). We also detected spontaneous Ca^{2+} spikes in GH3 cells expressing GeNL (Ca^{2+})₄₈₀ over 20 min. In contrast, the signals from Fura-2 were severely diminished 10 min after starting of observation due to phototoxic and photobleaching effects.”

b) Line 104, regarding "super resolution applications", it's not obvious how these reagents could be adapted for a super-resolution, especially of the type cited. I would suggest elaborating more fully or revising this out.

We apologize for our poor explanation.

Single-molecule-based superresolution imaging has been obtained with 'switching' between fluorescence on- and off-states that enable localization of individual molecules within a sub-diffraction area. Although this switching has been traditionally obtained by the use of photoswitchable proteins or organic dyes, “universal point accumulation imaging in the nanoscale topography” (uPAINT) technology employed the transient binding of fluorescently labeled tag to a sample. In the uPAINT approach, target molecules are individually imaged when a tagged fluorophore transiently binds to a target molecule, whereas unbound fluorophore is not detected due to fast diffusion. Here we demonstrated that the transient binding of his-tag GeNL to sparse Ni-NTAs provided the luminescent spots. Along this line, target molecules conjugated with Ni-NTA combined with his-tag GeNL might enable sub-diffraction imaging.

We have added the description to explain the above points in the revised manuscript as follows:

“The weak binding between GeNL and target molecules with Ni-NTA might enable sub-diffraction imaging, a similar manner to the “universal point accumulation imaging in the nanoscale topography” (uPAINT)¹⁷ method.”

c) legend, figure 2D, Which color is which sensor?

Thanks for pointing this out. We have revised the description in Figure 5 legend.

d) There is no discussion of the potential and unknown downsides to use of bioluminescence probes. Specifically, coelenterazine-like compounds are known for their antioxidant activity, which 'could' affect important biochemical pathways critical to cell function (see cardiomyocyte/XROS). The reader ought to be made aware of these potential issues, particularly because the use of these reagent are new and the effects of things like furimazine on cell function have not been fully characterized.

We thank the referee for helping to improve our manuscript. We have added the discussion about downsides to use of bioluminescent probes as follows:

“Second, the chemiluminescent substrates may affect cell behavior. Coelenterazine is reported to possess high anti-oxidant activity against reactive oxygen species²⁷ (ROS). Since furimazine is an analogue of coelenterazine, it might perturb cellular physiology by disruption of signal cascades involving ROS. To minimize the potential for this effect most of our experiments use <20 μ M furimazine, which does not affect cell viability and morphology⁴.”

e) I didn't note the presence of a section describing the cyan-colored reagent spectra, as was present for the other colors (e.g sup fig 4). Such information would be useful because the cyan spectra is really close to the luciferase.

We apologize for the oversight. The cyan variants picked in *E. coli* colonies were NOT

subjected to microplate reader screening on the basis of emission spectrum as the emission peaks of Nluc (donor, ~460 nm) and mTQ2 (acceptor, 480 nm) were too close to separate. The variants were directly purified and screened *in vitro* on the basis of brightness and BRET efficiency. The information has been added to the Methods as follow:

“Because the emission peaks of Nluc (donor, ~460 nm) and mTQ2 (acceptor, 480 nm) were too close to discern, the cyan-variants were directly purified and screened *in vitro* on the basis of brightness and BRET efficiency.”

Reviewer #2 (Remarks to the Author):

This is a well described development of a novel and highly original set of Nano-Lanterns that takes advantage of energy transfer from the bright bioluminescence protein Nanoluc to various hues of fluorescent proteins to enable multi-colour imaging without external excitation. I think these will be of wide interest to the community and open new avenues for experimentation with these unique indicators.

A substantial body of work has been presented which has been carefully undertaken and is of high quality. The simultaneous five-colour imaging and calcium indicator work is particularly impressive.

We are very grateful for Reviewer2's comment on our work.

A. The only element of the imaging that I didn't find compelling was the claim that a single clathrin-coated pit was identified in Figure 1e. The resolution is not good enough and the location seems to be predominantly perinuclear.

We have taken clearer images of clathrin coated pits by using a higher magnification objective lens ($\times 100$) to obtain better spatial resolution as suggested by the reviewer, the image of CCP's has been replaced in Figure 3 and change in magnification noted in the methods section.

Perinuclear localization of clathrin has previously been visualized by wide-field fluorescence microscopy, as shown below. The quoted image demonstrates punctate plasma membrane and perinuclear distribution (Ref. 18). A statement describing the perinuclear localization has been added to the results section to describe this issue with appropriate

referencing as follow:

“Notably, CeNL with a clathrin fusion tag demonstrated the perinuclear and punctate plasma membrane distribution corresponding to single clathrin-coated pits (CCPs), as described previously¹⁸.”

Figure 1d in Ref. 18

B. The figure legend to Figure 1f should also make clear that the outputs were submitted to spectral unmixing to separate out the colours.

We note the omission regarding use of spectral unmixing to separate out colors in the legend and have included a clear statement in the revision.

C. The single molecule imaging (Figure 1d) simply reflects imaging of individual particles that could represent aggregate complexes containing the GeNL lantern. How convinced are the authors that this represent single molecules of GeNL rather than single aggregates? If they are single molecules, does each detected particle have the same brightness?

We believe these are single molecules for several reasons. Firstly, calculation of photon number from single spots obtained the value 75 ± 30 photons (mean \pm SD, $n = 919$, Fig. 2). This is consistent with the predicted value estimated from the kinetic parameters associated with bulk solution analysis. Secondly, the trajectories of luminescence intensity at each ROI exhibited a stepwise transition between “Bright” states with 75 ± 30 photon emission and “Dark states” with emission similar to the background. We reasoned that those two states correspond to association and dissociation between Ni-NTA and single GeNL molecules

labeled with a his-tag, which might occur within the observation times.

As this matter is potentially controversial, as it is also raised by reviewer#3, to facilitate open discussion in the wider community we moved the relevant figure and note to the result section from Supplementary Information.

D. The manuscript is clearly written although doesn't reference all of the previous work with NanoLuc - which perhaps it should.

We appreciate this suggestion. Numerous reports cite the original NanoLuc paper (113 as of 19th August 2016), thus we have cited two comprehensive review papers in place of all original articles to facilitate a more general appeal of this approach

Saito, K. & Nagai, T. Recent progress in luminescent proteins development. *Curr Opin Chem Biol* **27**, 46-51 (2015).

(The paper summarizes recent advances in development of luminescent proteins, substrates, and indicators including NanoLuc.)

England, C.G., Ehlerding, E.B. & Cai, W. NanoLuc: A Small Luciferase Is Brightening Up the Field of Bioluminescence. *Bioconjug Chem* **27**, 1175-1187 (2016).

(The paper focuses on NanoLuc technology to review its versatile applications such as bioluminescence imaging and development of BRET-based biosensors e.t.c.)

E. Some discussion should be included regarding the requirements for the NanoLuc substrate and the need for it to equally access the Nano-lanterns in different cellular locations.

We thank the referee for helping to improve our manuscript. We have added to the discussion about the requirements for the substrate and its permeability into different cellular locations as follows:

“The luminescence signal decays over time by consumption of the luminescent substrate. This issue could be overcome by implementation of perfusion with a fresh luminescent substrate.” and “Third, coelenterazine and its analogues are reported to be a substrate for multidrug resistance (MDR1) P-glycoprotein transports²⁸. However, as we could easily detect

luminescence signals of eNLs in various cellular compartments, furimazine membrane permeability within the cells does not appear limiting.

F. It would be worth expanding Supplementary Table 2 to ensure that all of the Nanlanterns have the same K_m for furimazine.

We thank the referee for helping to improve our discussion regarding the characterization of eNLs. Accordingly we have measured the K_m , QY, k_{cat} of all eNLs and added this data and description to the Supplementary Figure 9, Table 2 and the result section, respectively, as follows:

“To investigate how the luminescence intensities of CeNL and GeNL became brighter than NLuc, we compared luminescence quantum yield (LQY) and the enzymatic parameters (K_m and k_{cat}) of the eNLs and NLuc (see **Supplementary Fig. 9 and Table 2**). k_{cat} of all eNLs were almost identical to that of NLuc, suggesting that FP fusion did not perturb NLuc enzymatic activity. In contrast, the LQY of CeNL and GeNL became larger than that of NLuc, while LQY of others were comparable or less (**Supplementary Table 2**). These results indicate that the enhancement of luminescence intensities in CeNL or GeNL are due to the enhancement of LQY by means of efficient BRET from NLuc to FPs with high fluorescence quantum yield.”

Reviewer #3 (Remarks to the Author):

This paper describes the development and experimental validation of five luminescent reporter constructs. The manuscript is well written and accessible, and the experimental methodology appropriate and rigorous. Nevertheless, there are some concerns regarding the utility and capabilities of these probes.

A. here is no doubt that these probes could be a useful addition to the imaging toolbox, however, the authors fail to demonstrate that they offer significant advantages over conventional fluorescent probes, as suggested (lines 36-39). There is no direct comparison between the new probes and conventional probes. Furthermore, the statement that long-term Ca^{2+} dynamics in iPS-derived cardiomyocytes is not feasible with fluorescent imaging (line 30) is not justified. Since Ca^{2+} imaging has been done in iPS cardiomyocytes, the authors need to demonstrate the advantage of their approach over conventional techniques before

making extraordinary claims.

Our reply to reviewer#1's similar comment (F) addresses this point with additional data, and text revision removing the previously over-stated claim. We hope this takes into account the view of reviewer#3 satisfactorily.

B. The claim of single molecule resolution is intriguing (line 101-104), however, based on the evidence provided in Fig 1d, Supp Fig 10 and Note 2, this may be more speculative than the authors suggest. Given that single molecule imaging could be an important application for the probe then the claim should be better supported and included in the main body of the text, not as a supplementary note.

We appreciate the reviewer raising this point. Reviewer#2 gave similar comments. We hope our response to comment C of reviewer#2 addresses the concerns of reviewer#3.

Reviewer #1 (Remarks to the Author):

This is a resubmission of an interesting manuscript describing multicolored bioluminescence reporters. The authors have more than adequately addressed the minor concerns raised in the previous round of reviews, which focused on the nuances of presentation. The substance of the paper remains truly exciting! No further comments.

Reviewer #2 (Remarks to the Author):

I am happy that the authors have adequately dealt with the concerns that I raised previously.

Reviewer #3 (Remarks to the Author):

The authors have fully addressed all concerns.